# A Bounding Box is Worth One Token: Interleaving Layout and Text in a Large Language Model for Document Understanding

## Abstract

Recently, many studies have demonstrated that exclusively incorporating OCR-derived text and spatial layouts with large language models (LLMs) can be highly effective for document understanding tasks. However, existing methods that integrate spatial layouts with text have limitations, such as producing overly long text sequences or failing to fully leverage the autoregressive traits of LLMs. In this work, we introduce *Interleaving **Lay**out and **Text** in a Large Language Model (LayTextLLM)* for document understanding. In particular, LayTextLLM projects each bounding box to a single embedding and interleaves it with text, efficiently avoiding long sequence issues while leveraging autoregressive traits of LLMs. LayTextLLM not only streamlines the interaction of layout and textual data but also shows enhanced performance in Key Information Extraction (KIE) and Visual Question Answering (VQA). Comprehensive benchmark evaluations reveal significant improvements, with a 27.0% increase on KIE tasks and 24.1% on VQA tasks compared to previous state-of-the-art document understanding MLLMs, as well as a 15.5% improvement over other SOTA OCR-based LLMs on KIE tasks.

## 1 Introduction

Recent research has increasingly focused on applying Large Language Models (LLMs) [1–17] to document-oriented Visual Question Answering (VQA) and Key Information Extraction (KIE) scenarios. Efforts to build a text-sensitive MultiModal Large Language Models (MLLMs) based on existing LLMs, particularly aimed at enhancing Visually Rich Document Understanding (VRDU), have made significant progress [6, 12, 18]. Although existing MLLMs show promising results in document understanding, they often encounter challenges related to image resolution. When the input image is of low resolution, it is too blurry to extract visual features effectively. Conversely, high-resolution images require additional computational resources to capture detailed textual information [12].

Concurrently, another line of research employs off-the-shelf OCR tools to extract text and spatial layouts, which are then combined with LLMs to address VRDU tasks. These approaches assume that *most valuable information for document comprehension can be derived from the text and its spatial layouts, viewing spatial layouts as "lightweight visual information" [19]*. Following this premise, several studies [12, 20–23] have explored various approaches that integrate spatial layouts with text for LLMs, achieving results that are competitive with, or even surpass, those of MLLMs.

The most natural method to incorporate layout information is by treating spatial layouts as tokens, which allows for the seamless interleaving of text and layout into a unified text sequence [20, 22, 23]. For example, Perot et al. [20] employ format such as *"HARRISBURG 78|09"* to represent OCR text and corresponding layout, where *"HARRISBURG"* is OCR text and *"78|09"* indicates the mean of

the horizontal and vertical coordinates, respectively. Similarly, He et al. [23] use *"[x_min, y_min, x_max, y_max]"* to represent layout information. These approaches can effectively take advantage of autoregressive characteristics of LLMs and is known as the *"coordinate-as-tokens"* scheme [20]. In contrast, DocLLM [19] explores interacting spatial layouts with text through a disentangled spatial attention mechanism that captures cross-alignment between text and layout modalities.

However, we argue that both of the previous approaches have limitations. As shown in Fig. 1, coordinate-as-tokens significantly increases the number of tokens. Additionally, to accurately comprehend coordinates and enhance zero-shot capabilities, this scheme often requires few-shot in-context demonstrations and large-scale language models, such as ChatGPT Davinci-003 (175B) [23], which exacerbates issues related to sequence length and GPU resource demands. Meanwhile, although DocLLM does not increase sequence length and integrates spatial layouts through attention, its generalizability is limited. We believe that spatial cross attention and masked span tasks in DocLLM cannot fully utilize the autoregressive traits of LLMs.

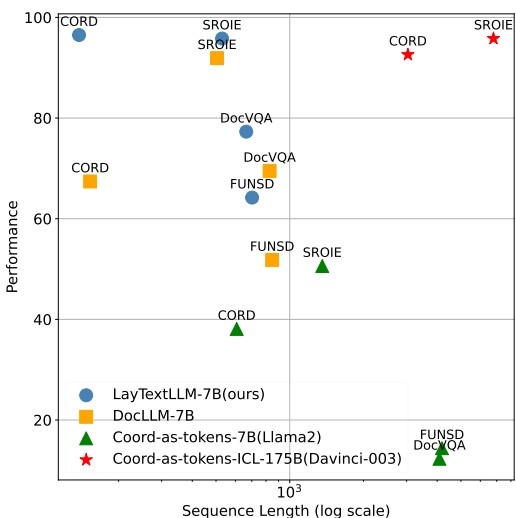

Figure 1: The performance against input sequence length of different datasets across various OCR-based methods where data is from Tab. 2 and 5.

To address these problems, this paper explores a simple yet effective approach to enhance the interaction between spatial layouts and text — *Interleaving **Lay**out and **Text** in a **L**arge **L**anguage **M**odel (LayTextLLM)* for document understanding. Adhering to the common practice of interleaving any modality with text [15, 24, 25], we specifically apply this principle to spatial layouts. In particular, we maps each bounding box to a single embedding, which is then interleaved with its corresponding text. Then we propose a tailored pre-training task—Layout-aware Next Token Prediction—a completely self-supervised task that enhances the alignment between layout and textual modalities without using synthetic data. Finally, through the proposed Shuffled-OCR Supervised Fine-tuning, LayTextLLM significantly improves performance on downstream document-related VQA and KIE tasks. As shown in Fig. 1, LayTextLLM significantly outperforms the 175B models, while only slightly increasing or even reducing the sequence length compared to DocLLM. Our contributions can be listed as follows:

- We propose LayTextLLM for document understanding. To the best of the authors' knowledge, this is the first work to employ a unified embedding approach (Sec. 3.1.1) that interleaves spatial layouts directly with textual data within a LLM. By representing each bounding box with one token, LayTextLLM efficiently addresses sequence length issues brought by coordiante-as-tokens while fully leveraging autoregressive traits for enhanced document understanding.

- We propose two tailored training tasks: (1) Layout-aware Next Token Prediction (Sec. 3.2.1), a completely self-supervised training task to enhance the alignment between layout and textual modality; (2) Shuffled-OCR Supervised Fine-tuning task (Sec. 3.2.2) to better elicit the model generalizability in downstream tasks.

- Comprehensive experimental results demonstrate quantitatively that LayTextLLM significantly outperforms previous state-of-the-art (SOTA) OCR-free MLLMs by a large margin in zero-shot scenarios, particularly in KIE tasks with an improvement of 27.0%. Additionally, we illustrate that LayTextLLM competes effectively or even surpasses previous SOTA OCR-based methods in both zero-shot and SFT scenarios. Specifically, it surpasses DocLLM by 19.8% on VQA and 15.5% on KIE tasks (Sec. 4).

- Extensive ablations demonstrate the utility of the proposed component, with analysis showing that LayTextLLM not only improves performance but also reduces input sequence length compared to current OCR-based models.

## 2 Related Work

### 2.1 OCR-based LLMs for Document Understanding

Early document understanding methods [26–30] tend to solve the task in a two-stage manner, *i.e.*, first reading texts from input document images using off-the-shelf OCR engines and then understanding the extracted texts. Considering the advantages of LLMs (*e.g.*, high generalizability), some recent methods endeavor to combine LLMs with OCR-derived results to solve document understanding. For example, inspired by the "coordinate-as-tokens" scheme [20], He et al. [23] propose to use *"[x_min, y_min, x_max, y_max]"* to introduce the layout information, which can fuse the layout information and texts into a unified text sequence and fully exploit the autoregressive merit of LLMs. To reinforce the layout information while avoiding increasing the number of tokens, DocLLM [19] designs a disentangled spatial attention mechanism to capture cross-alignment between text and layout modalities. Recently, LayoutLLM [21] utilizes the pre-trained layout-aware model [31], to insert the visual information, layout information and text information. However, the aforementioned methods neither suffer from the computational overhead leading by the increasing tokens or hardly take advantage of autoregressive characteristics of LLMs. Thus, it is an urgent problem to address how to better incorporate layout information without significantly increasing the number of tokens.

### 2.2 OCR-free MLLMs for Document Understanding

Another approach to solve document understanding tasks is the OCR-free method. Benefiting from the end-to-end training framework, it involves processing the text content of documents directly, without relying on OCR engines. Donut [32] first presents an OCR-free method through mapping a text-rich document image into the desired answers. Pix2Struct [33] is trained to parse masked screenshots of web pages into simplified HTML, where variable resolution inputs are supported. While these approaches eliminate the need for OCR tools, they still necessitate task-specific fine-tuning. With the increasing popularity of LLMs/MLLMs [10–17], various methods are proposed to solve the document understanding task through explicitly training models on visual text understanding datasets and fine-tuning them with instructions to perform a zero-shot prediction. LLaVAR [34] and UniDoc [10] are notable examples that expand upon the document-oriented VQA capabilities of LLaVA [35] by incorporating document-based tasks. These models pioneer the use of MLLMs for predicting texts and coordinates from document images, enabling the development of OCR-free document understanding methods. Additionally, DocPedia [9] operates document images in the frequency domain, allowing for higher input resolution without increasing the input sequence length. Recent advancements in this field, including mPLUG-DocOwl [18], Qwen-VL [6], and TextMonkey [12], leverage publicly available document-related VQA datasets to further enhance the document understanding capability. Although these OCR-free methods have exhibited their advantages, they still struggle with the high-resolution input to reserve more text-related details.

## 3 Method

In this section, we present our LayTextLLM. First, we introduce a innovative Spatial Layout Projector (Sec. 3.1.1) converts four-dimensional layout coordinates into a single-token embedding. To reduce parameter overhead, we apply Partial Low-Rank Adaptation (Sec. 3.1.2). We also introduce two specific training tasks: Layout-aware Next Token Prediction (Sec. 3.2.1) to align layouts with text during pre-training, and Shuffled-OCR Supervised Fine-tuning (Sec. 3.2.2) to enhance the generalizability of the model. An illustration of our approach is shown in Fig. 2.

### 3.1 Model Architecture

LayTextLLM is built on the Llama2-7B-base model, which was originally designed to accept only text inputs [36, 37]. To enable the model to interleave spatial layouts with text, we introduce a novel Spatial Layout Projector. This projector converts OCR-derived coordinates into bounding box tokens. We also adopt the Partial Low-Rank Adaptation, a minimally invasive method to incorporate additional modalities while preserving the LLM's inherent knowledge intact.

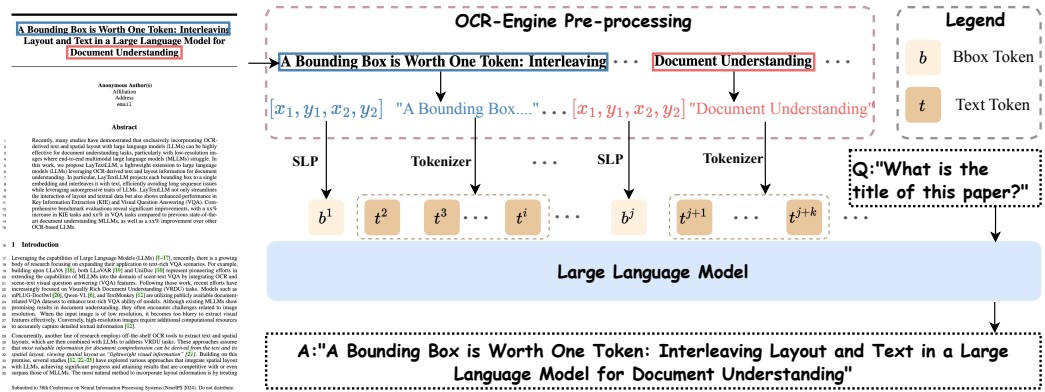

Figure 2: An overview of LayTextLLM incorporates interleaving bounding box tokens ($b^i$) with text tokens ($t^i$), where the superscripts represent the sequence positions of the tokens.

### 3.1.1 Spatial Layout Projector (SLP)

A key innovation in LayTextLLM is the Spatial Layout Projector (SLP), which transforms a spatial layout into a singular bounding box token. This enhancement enables the model to process both spatial layouts and textual inputs simultaneously. To be specifically, each OCR-derived spatial layout is represented by a bounding box defined by four-dimensional coordinates $[x_1, y_1, x_2, y_2]$, these coordinates represent the normalized minimum and maximum horizontal and vertical extents of the box, respectively. The SLP maps these coordinates into a high-dimensional space that the language model can process as a single token. The process can be computed as $z = W \cdot c + b$, where $c \in \mathbb{R}^4$ is the vector of the bounding box coordinates. $W \in \mathbb{R}^{d \times 4}$ is a weight matrix with $d$ represents the dimension of the embedding, $b \in \mathbb{R}^{d \times 1}$ is a bias vector, $z$ is the resulting bounding box token represented as an $d$-dimensional embedding. As illustrated in Fig. 2, the resulting bounding box token $z$ will be interleaved with corresponding textual embeddings to put into LLMs. Note that the SLP is shared by all bounding box tokens so very limited number of parameters are introduced.

Compared to the coordinate-as-tokens scheme, the SLP represents each bounding box with a single token. This approach significantly reduces the number of input tokens and adheres to the practice of interleaving any modality with text, effectively integrating layout and textual information into a unified sequence. This allows the model to process both modalities simultaneously and coherently, fully leveraging the autoregressive traits of LLMs.

### 3.1.2 Layout Partial Low-Rank Adaptation

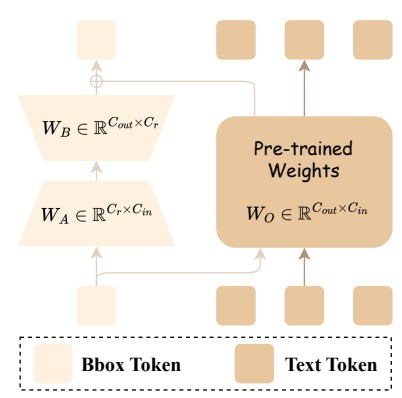

Figure 3: The illustration of P-LoRA, adapted from [15].

After using the SLP to generate bounding box tokens and a tokenizer to produce text tokens, these two modalities are then communicated using a Layout Partial Low-Rank Adaptation (P-LoRA) module in LLMs. P-LoRA, introduced in InternLM-XComposer2 [15], is originally used to adapt LLMs to visual modality. It applies plug-in low-rank modules specified to the visual tokens, which adds minimal parameters while preserving the LLMs inherent knowledge.

Formally, as shown in Fig. 3 for a linear layer in the LLM, the original weights $W_O \in \mathbb{R}^{C_{out} \times C_{in}}$ and bias $B_O \in \mathbb{R}^{C_{out}}$ are specified for input and output dimensions $C_{in}$ and $C_{out}$. P-LoRA modifies this setup by incorporating two additional matrices, $W_A \in \mathbb{R}^{C_r \times C_{in}}$ and $W_B \in \mathbb{R}^{C_{out} \times C_r}$. These matrices are lower-rank, with $C_r$ being considerably smaller than both $C_{in}$ and $C_{out}$, and are specifically designed to interact with new modality tokens, which in our case are bounding box tokens. For example, given an input $x =$

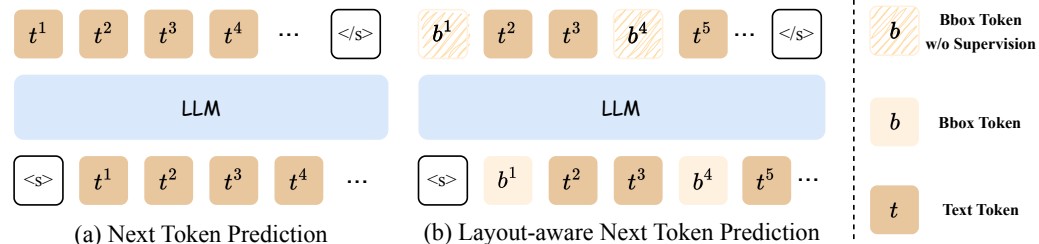

Figure 4: Comparison of Layout-aware Next Token Prediction and normal Next Token Prediction.

$[x_b, x_t]$ comprising of bounding box tokens $(x_b)$ and textual tokens $(x_t)$ is fed into the system, the forward process is as follows, where $\hat{x}_t, \hat{x}_b$ and $\hat{x}$ are outputs:

$$\begin{aligned}
\hat{x}_t &= W_0 x_t + B_0 \\
\hat{x}_b &= W_0 x_b + W_B W_A x_b + B_0 \\
\hat{x} &= [\hat{x}_b, \hat{x}_t]
\end{aligned} \tag{1}$$

## 3.2 Training Procedure

LayTextLLM is trained with innovative layout-aware training procedure, which consists of two stages: Layout-aware Next Token Prediction pre-training and Shuffled-OCR Supervised Fine-tuning.

### 3.2.1 Layout-aware Next Token Prediction

Inspired by the next token prediction commonly used in current LLM pre-training [1–7], we propose the Layout-aware Next Token Prediction (LNTP). Fig. 4 presents the contrast of the proposed Layout-aware Next Token Prediction and the conventional next token prediction task. The traditional next token prediction (Fig. 4(a)) relies solely on the textual content, predicting each subsequent token based on the prior sequence of tokens without considering their spatial layouts. Layout-aware next token prediction (Fig. 4(b)), however, interleaves the spatial information encoded by SLP (*i.e.*, $b^i$) with the text tokens (*i.e.*, $t^i$). This integration considers both the content and its layout within the document, leading to a richer, more precise understanding of both the structure and the content.

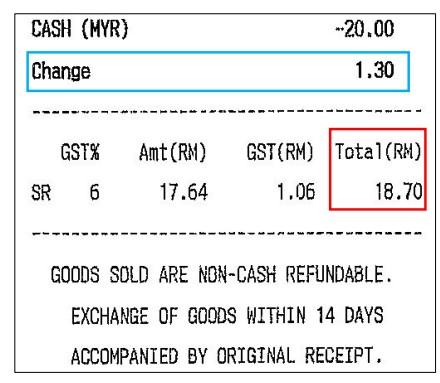

Figure 5: Receipt layout example.

Similarly, primary objective of LNTP is to maximize the likelihood of its predictions for the next token. Thus the loss function is defined as

$$\mathcal{L} = -\frac{1}{T} \sum_{i=1}^{T} \log P\left(t^i \mid t^1, t^2, \ldots, t^{i-1}\right) \tag{2}$$

where $P\left(t^i \mid t^1, t^2, \ldots, t^{i-1}\right)$ represents the probability of $i^{th}$ token $t^i$ given the sequence of preceding tokens $t^1, t^2, \ldots, t^{i-1}$, as predicted by the model. Note that we compute the loss only for text tokens, excluding bounding box tokens. During pre-training, our goal is to enhance the alignment between spatial layouts and textual modality, while preserving the LLM's inherent knowledge as much as possible. Thus, we freeze the LLMs and only update the parameters of SLP and P-LoRA.

It is important to note that the proposed Layout-aware Next Token Prediction is a completely self-supervised pre-training procedure, unlike previous works that require human annotations of document structure data or synthetic data generated by larger LLMs such as GPT-4 [21]. Thus, LNTP facilitates the creation of large-scale, high-fidelity pre-training datasets at minimal cost.

### 3.2.2 Shuffled-OCR Supervised Fine-tuning

OCR engines typically process text from top to bottom and left to right. This order is also adopted as the input sequence for current OCR-based LLMs [19, 21]. However, modern LLMs often exhibit a strong inductive bias toward the positions of input tokens, influenced by designs such as Rotary Position Embeddings (RoPE) [38]. Specifically, tokens that are close together in the input sequence are likely to receive higher attention scores, which is advantageous for processing standard text sequences. Such inductive bias brings cons and pros.

Consider the example illustrated in Fig. 5, where the OCR input text reads: *" ... Change, 1.30, GST%, Amt(RM), GST(RM), Total(RM), SR, 6, 17.64, 1.06, 18.70 ... "*. If the question posed is *"What is the value of the field Change?"* (highlighted in a blue box), the model easily identifies *"1.30"* as it is closely positioned to the word *"Change"* in the sequence. However, for a more challenging query like *"What is the value of the field Total(RM)?"* (highlighted in a red box), the model struggles to determine the correct answer due to the presence of multiple subsequent numbers closed to *"Total(RM)"*. LayTextLLM integrates spatial layouts with textual data, reducing reliance on input sequence order. Thus, we posit that shuffling the OCR input order could enhance the resilience of LayTextLLM in discerning relevant information irrespective of token proximity in the sequence.

Specifically, we propose Shuffled-OCR Supervised Fine-tuning (SSFT) that randomly shuffles the order of OCR-derived text in a certain proportion of examples. The range of exploration for the shuffling ratio can be found in Tab. 7 and 20% shuffled ratio is applied. The training objective is equivalent to predicting the next tokens, but in this scenario, only the tokens of the response are used to compute loss. During SSFT, we unfreeze all parameters including those of LLMs. Experimental results in Section 4.6 demonstrate that utilizing SSFT can further enhance model performance, making it more robust to disruptions in input token order.

## 4 Experiments

### 4.1 Datasets

**Pre-training data**    In our training process, we exclusively use open-source data to facilitate replication. We collect data from two datasets for pre-training: (1) **IIT-CDIP Test Collection 1.0** [39] and (2) **DocBank** [40]. The IIT-CDIP Test Collection 1.0 comprises an extensive repository of more than 16 million document pages. DocBank consists of 500K documents, each presenting distinct layouts with a single page per document. For training efficiency, we choose to utilize the entire DocBank dataset and only subsample 5 million pages from the IIT-CDIP collection 1.0.

**SFT data**    For document-oriented VQA, we select **Document Dense Description (DDD)** and **Layout-aware SFT** data used in Luo et al. [21], which are two synthetic datasets generated by GPT-4. Besides, **DocVQA** [41], **InfoVQA** [42], **ChartQA** [43], **VisualMRC** [44] is included following [12]. For KIE task, we select **SROIE** [45], **CORD** [46], **FUNSD** [47], **POIE** [48] datasets following [12, 19, 21].

### 4.2 Implementation Detail

The LLM component of LayTextLLM is initialized from the Llama2-7B-base [36], which is a widely-used backbone. Other parameters including SLP and P-LoRA are randomly initialized. During pre-training, the LLM is frozen, and the parameters of SLP and P-LoRA modules are updated. During SFT, all parameters are fine-tuned. Other detailed setup can be found in Appendix B.

We have configured the model with three versions of LayTextLLM for a side-by-side comparison under different settings. Aligned with Luo et al. [21], the first version, **LayTextLLM**$_{zero}$, is trained exclusively with DDD and Layout-aware SFT data. Building upon this, and in alignment with the setting of Liu et al. [12], we introduce the DocVQA, InfoVQA, and ChartQA training sets to the dataset pool for our second version, termed **LayTextLLM**$_{vqa}$. Finally, we incorporate a comprehensive suite of KIE datasets—FUNSD, CORD, POIE, SROIE, and VisualMRC—as described by Wang et al. [19], creating our most extensive version, **LayTextLLM**$_{all}$. Note that all versions are based on the same pre-trained LayTextLLM weight.

| | Document-Oriented VQA | | | KIE | | | |
|---|---|---|---|---|---|---|---|
| | DocVQA | InfoVQA | Avg | FUNSD | SROIE | POIE | Avg |
| **Metric** | *Accuracy %* | | | | | | |
| **OCR-free** | | | | | | | |
| UniDoc [10] | 7.7 | 14.7 | 11.2 | 1.0 | 2.9 | 5.1 | 3.0 |
| DocPedia [9] | $47.1^*$ | $15.2^*$ | 31.2 | 29.9 | 21.4 | 39.9 | 30.4 |
| Monkey [49] | $50.1^*$ | $25.8^*$ | 38.0 | 24.1 | 41.9 | 19.9 | 28.6 |
| InternVL [50] | $28.7^*$ | $23.6^*$ | 26.2 | 6.5 | 26.4 | 25.9 | 19.6 |
| InternLM-XComposer2 [15] | 39.7 | 28.6 | 34.2 | 15.3 | 34.2 | 49.3 | 32.9 |
| TextMonkey [12] | $64.3^*$ | $28.2^*$ | 46.3 | 32.3 | 47.0 | 27.9 | 35.7 |
| TextMonkey$_+$ [12] | $66.7^*$ | $28.6^*$ | 47.7 | 42.9 | 46.2 | 32.0 | 40.4 |
| **text + polys** | | | | | | | |
| LayTextLLM$_{zero}$ (Ours) | 71.8 | 33.8 | 52.8 | **49.4** | **86.7** | 66.1 | **67.4** |
| LayTextLLM$_{vqa}$ (Ours) | $\mathbf{77.4}^*$ | $\mathbf{66.1}^*$ | **71.8** | 48.9 | 74.6 | **70.6** | 64.7 |

Table 1: Comparison with SOTA OCR-free MLLMs. $^*$ indicates the training set used.

| | Document-Oriented VQA | | | KIE | | | |
|---|---|---|---|---|---|---|---|
| | DocVQA | VisualMRC | Avg | FUNSD | CORD | SROIE | Avg |
| **Metric** | *ANLS % / CIDEr* | | | *F-score %* | | | |
| **Text** | | | | | | | |
| Llama2-7B-base | 34.0 | 182.7 | 108.3 | 25.6 | 51.9 | 43.4 | 40.3 |
| Llama2-7B-chat | 20.5 | 6.3 | 13.4 | 23.4 | 51.8 | 58.6 | 44.6 |
| **Text + Polys** | | | | | | | |
| Llama2-7B-base$_{coor}$ [23] | 8.4 | 3.8 | 6.1 | 6.0 | 46.4 | 34.7 | 29.0 |
| Llama2-7B-chat$_{coor}$ [23] | 12.3 | 28.0 | 20.1 | 14.4 | 38.1 | 50.6 | 34.3 |
| Davinci-003-175B$_{coor}$ [23] | - | - | - | - | 92.6 | 95.8 | - |
| DocLLM [19] | $69.5^*$ | $264.1^*$ | 166.8 | $51.8^*$ | $67.4^*$ | $91.9^*$ | 70.3 |
| LayTextLLM$_{zero}$ (Ours) | 65.4 | 260.7 | 163.0 | 48.6 | 74.5 | 86.4 | 69.8 |
| LayTextLLM$_{vqa}$ (Ours) | $75.7^*$ | $260.2^*$ | 168.0 | 52.7 | 70.9 | 78.6 | 67.4 |
| LayTextLLM$_{all}$ (Ours) | $\mathbf{77.3}^*$ | $\mathbf{295.9}^*$ | **186.6** | $\mathbf{64.2}^*$ | $\mathbf{96.5}^*$ | $\mathbf{95.8}^*$ | **85.8** |

Table 2: Comparison with other OCR-based methods. $^*$ indicates the training set used.

## 4.3 Baselines

**OCR-free baselines** In the category of OCR-free MLLMs, we have chosen the following SOTA models as our strong baselines due to their superior performance in both document-oriented VQA and KIE tasks. These include **UniDoc** [10], **DocPedia** [9], **Monkey** [49], **InternVL** [50], **InternLM-XComposer2** [15], **TextMonkey**, and **TextMonkey$_+$** [12].

**OCR-based baselines** For OCR-based baseline models, we implemented a basic approach using only OCR-derived text as input. This was done using two versions: **Llama2-7B-base** and **Llama2-7B-chat**. We also adapted the coordinate-as-tokens scheme from He et al. [23] for these models, resulting in two new variants: **Llama2-7B-base$_{coor}$** and **Llama2-7B-chat$_{coor}$**. It's important to note that we did not employ the ICL strategy with these models, as it would significantly exceed their maximum sequence length constraints. Additionally, we included results from a stronger baseline using the ChatGPT Davinci-003 (175B) model [23], termed **Davinci-003-175B$_{coor}$**. One other recent SOTA OCR-based approach, **DocLLM** [19] is also considered in our analysis. Finally, **LayoutLLM** and **LayoutLLM$_{CoT}$** [21], which integrates visual cues, text and layout is also included.

## 4.4 Evaluation Metrics

To ensure a fair comparison with OCR-free methods, we adopted the accuracy metric, where a response from the model is considered correct if it fully captures the ground truth. This approach aligns with the evaluation criteria described by [9, 10, 12]. To further enhance the comparability with other OCR-based methods, we conducted additional evaluations using original metrics specific to certain datasets, such as F1 score [19, 23], ANLS [19, 21, 51] and CIDEr [19, 52].

| | Document-Oriented VQA | | | KIE | | | |
| | DocVQA | VisualMRC | Avg | FUNSD[-] | CORD[-] | SROIE[-] | Avg |
|---|---|---|---|---|---|---|---|
| **Metric** | *ANLS %* | | | | | | |
| **Visual + Text + Polys** | | | | | | | |
| LayoutLLM [21] | 72.3 | - | - | 74.0 | - | - | - |
| LayoutLLM$_{CoT}$ [21] | 74.2 | **55.7** | **64.9** | 79.9 | 63.1 | 72.1 | 71.7 |
| **Text** | | | | | | | |
| Llama2-7B-base | 34.0 | 25.4 | 29.7 | 42.1 | 46.7 | 60.6 | 49.8 |
| Llama2-7B-chat | 20.5 | 9.9 | 15.2 | 15.1 | 20.0 | 35.6 | 23.5 |
| **Text + Polys** | | | | | | | |
| Llama2-7B-base$_{coor}$ [23] | 8.4 | 6.7 | 7.5 | 4.3 | 33.0 | 47.2 | 28.1 |
| Llama2-7B-chat$_{coor}$ [23] | 12.3 | 12.2 | 12.2 | 11.9 | 6.4 | 39.4 | 19.2 |
| LayTextLLM$_{zero}$ (Ours) | 65.4 | 36.2 | 50.8 | 71.0 | 66.9 | 89.2 | 75.7 |
| LayTextLLM$_{all}$ (Ours) | **77.3**[*] | **41.7**[*] | 59.5 | **81.3**[*] | **82.6**[*] | **96.2**[*] | **86.7** |

Table 3: Comparison with LayoutLLM. [-] indicates that the cleaned test set used in Luo et al. [21].

## 4.5 Quantitative Results

**Comparison with SOTA OCR-free Methods**    The experimental results shown in Tab. 1 demonstrate the outstanding performance of the LayTextLLM series across various tasks. Note that the results for ChartQA are reported in Appendix E due to concerns about fairness in comparison, as the dataset does not include OCR-derived results and we used in-house OCR tools instead. Firstly, LayTextLLM$_{zero}$ significantly outperforms previous SOTA OCR-free methods, such as TextMonkey [12], in zero-shot capabilities, even when these methods use the training set of the dataset. For example, in the DocVQA and InfoVQA datasets, LayTextLLM$_{zero}$ achieves accuracies of 71.8% and 33.8%, respectively, which are markedly higher than existing OCR-free methods such as TextMonkey and InternLM-XComposer2. When fine-tuned with corresponding datasets, LayTextLLM shows even greater performance improvements, particularly in document-oriented VQA datasets. Specifically, its accuracies on DocVQA and InfoVQA increase to 77.4% and 66.1%, respectively, demonstrating the model's strong ability to leverage task-specific data. Additionally, LayTextLLM$_{zero}$ excels in KIE datasets, particularly on the SROIE and POIE datasets, achieving accuracies of 86.7% and 66.1%, respectively. These results significantly surpass those of previous SOTA OCR-free model (*i.e.*, TextMonkey$_+$) by margins of 40.5% and 34.1%, respectively. This significant performance gain is likely due to these datasets containing low-resolution images that are too blurred for current MLLMs to extract visual features, whereas LayTextLLM shows robustness in such challenging scenarios.

**Comparison with SOTA OCR-based Methods**    For comprehensive comparison, we have also conducted corresponding experiments to align with OCR-based methods [19, 21]. The experimental results presented in Tab. 2 showcase significant performance improvements achieved by LayTextLLM models compared to pure OCR-based SOTA methods such as DocLLM [19]. Specifically, when comparing with DocLLM, LayTextLLM$_{zero}$ demonstrates notably superior performance, with even its zero-shot capabilities being competitive with supervised SFT approaches. We believe that the subpar performance of DocLLM is likely due to its use of cross-attention and the masked span pre-training tasks [53], which fail to leverage the autoregressive features of LLMs effectively. Similarly, when contrasting with coordinate-as-tokens employed in Llama2-7B, LayTextLLM$_{zero}$ again outperforms significantly. This disparity in performance can be attributed to the following three reasons: (1) The coordinate-as-tokens approach tends to introduce an excessive number of tokens, often exceeding the pre-defined maximum length of Llama2-7B (*i.e.*, 4096). Consequently, this leads to a lack of crucial OCR information, resulting in hallucination and subpar performance. (2) When re-implementing the coordinate-as-tokens method with Llama2-7B, we did not introduce the ICL strategy, as it would contribute additional length to the input sequence. (3) The coordinate-as-tokens approach necessitates a considerably larger-sized LLM to comprehend the numerical tokens effectively.

In comparison to LayoutLLM [21], our approach exhibits discrepant performance in different tasks, as shown in Tab. 3. In zero-shot scenarios, we outperform LayoutLLM in most KIE datasets, validating our capability to leverage OCR-based results effectively. However, we fall short on document-oriented VQA tasks since answering some questions that are strongly related to vision information may challenge our approach. Two main reasons may well explain this performance discrepancy:

(1) The visual encoder in LayoutLLM provides additional visual information. (2) LayoutLLM incorporates the Chain-of-Thought (CoT) mechanism to model contextual information while it is not used in our approach. However, when fine-tuned with tailored data, LayTextLLM significantly outperforms LayoutLLM, showcasing its strong ability to utilize task-specific data. More qualitative example demonstrates can be found in Appendix A.

## 4.6   Analysis

| LNTP | SSFT | Document-Oriented VQA | | | | KIE | | | | |
|---|---|---|---|---|---|---|---|---|---|---|
| | | DocVQA | InfoVQA | VisualMRC | Avg | FUNSD | CORD | SROIE | POIE | Avg |
| | ✓ | 72.3 | 35.7 | 24.4 | 44.2 | 50.6 | 91.6 | 92.8 | 58.6 | 73.4 |
| ✓ | | 74.5 | 38.0 | 23.9 | 45.5 | 56.4 | 95.8 | 93.2 | 59.6 | 76.3 |
| ✓ | ✓ | **78.8** | **42.7** | **35.1** | **52.2** | **62.9** | **95.9** | **95.2** | **61.7** | **78.9** |

Table 4: Ablations on pre-training and SFT component of LayTextLLM (Accuracy).

**Ablations**   To better assess the utility of Layout-aware Next Token Prediction and Shuffled-OCR Supervised Fine-tuning in LayTextLLM, an ablation study was performed (see Tab. 4). Details on the training setup for all variants are provided in Appendix B. It is evident that both LNTP and SSFT significantly enhance the utility of LayTextLLM. Specifically, disabling LNTP results in an 8% decrease in performance on VQA tasks and a 5.5% decrease on KIE tasks. Disabling SSFT leads to a decrease in average accuracy by 6.7% and 2.6% for VQA and KIE tasks, respectively.

**Sequence Length**   Tab. 5 presents statistics on the average input sequence length across different datasets. Intriguingly, despite interleaving bounding box tokens, LayTextLLM consistently exhibits the shortest sequence length in three out of four datasets, even surpassing DocLLM, which is counterintuitive. We attribute this to the tokenizer mechanism. For example, using tokenizer.encode(), a single word from the OCR engine, like *"International"* is encoded into a single ID [4623]. Conversely, when the entire OCR output is processed as one sequence, such as *"... CPC,International,Inc..."*, the word *"International"* is split into two IDs [17579, 1288], corresponding to *"Intern"* and *"ational"* respectively. This type of case occurs frequently, more discussion in Appendix C.

| Dataset | LayTextLLM | DocLLM [19] | Coor-as-tokens [23] |
|---|---|---|---|
| DocVQA | **664.3** | 827.5 | 4085.7 |
| CORD | **137.9** | 153.2 | 607.3 |
| FUNSD | **701.9** | 847.5 | 4183.4 |
| SROIE | 529.2 | **505.1** | 1357.7 |

Table 5: Average sequence length of each data for different methods using Llama2 tokenizer.

## 5   Limitation

Although LayTextLLM has shown significant capabilities in text-rich VQA and KIE tasks, this alone does not suffice for all real-world applications. There are some instances, particularly in chart analysis, where reasoning must be based solely on visual cues (*e.g.* size, color)—a challenge that remains unmet. Questions such as *"What is the difference between the highest and the lowest green bar?"* illustrate this gap. The ChartQA results, detailed in Appendix E, also underscore these limitations. Addressing these challenges highlights the urgent need for future enhancements that integrate visual cue within the capabilities of LayTextLLM.

## 6   Conclusion

We propose LayTextLLM for various VRDU tasks, in which spatial layouts and textual data are seamlessly interleaved to make more accurate prediction by introducing a innovative Spatial Layout Projector. Two tailored training tasks — Layout-aware Next Token Prediction and Shuffled-OCR Supervised Fine-tuning — are designed to improve the comprehension of document layouts. Extensive experiments confirm the effectiveness of LayTextLLM.

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

# Appendix

# A   Qualitative Examples

Qualitative examples of document-oriented VQA (upper row) and KIE (bottom row) are shown in Fig. 6. The results indicate that LayTextLLM is highly effective in utilizing spatial layout information to make more accurate predictions for these challenging examples. For example, in the upper right figure, many numeric texts in the receipt act as noise for the baseline method. In contrast, LayTextLLM integrates layout information to accurately predict the total price, as demonstrated by the other examples, underscoring the utility of LayTextLLM.

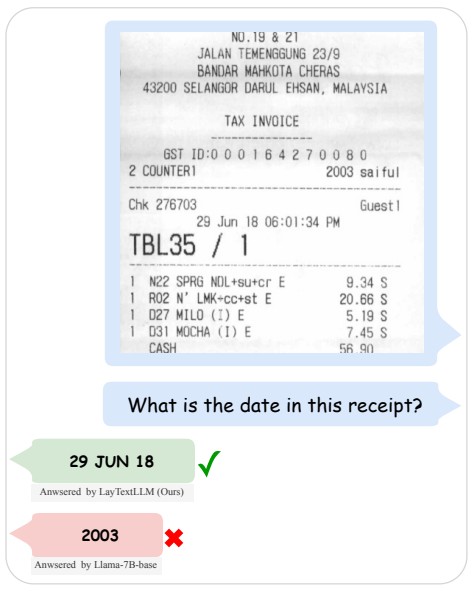
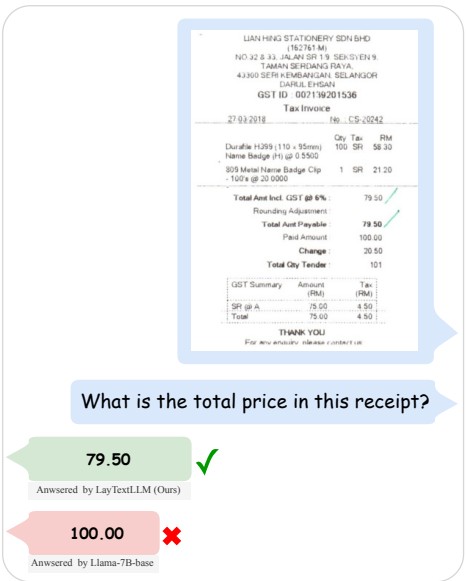
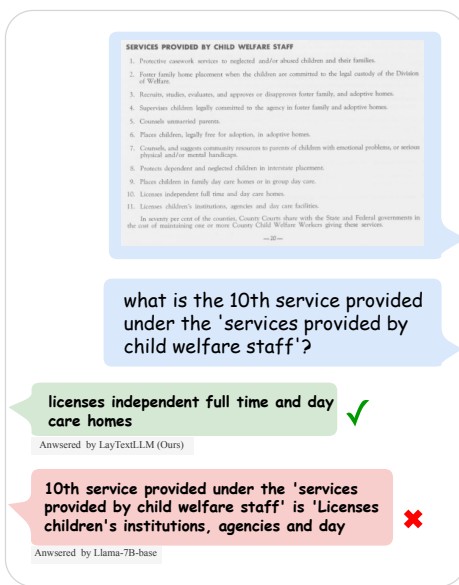
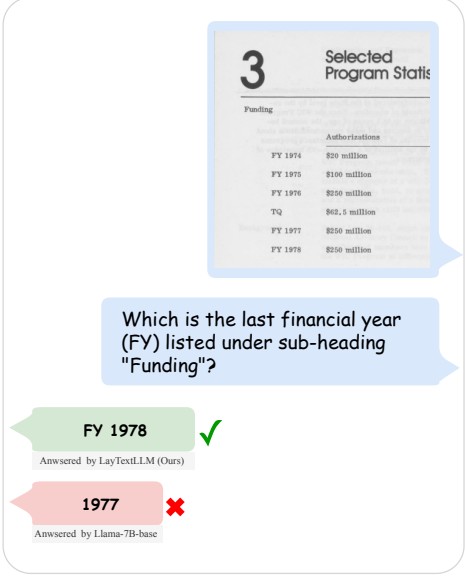

Figure 6: Qualitative comparison with the baseline method.

## B  Implementation Detail

All training and inference procedures are conducted on eight NVIDIA A100 GPUs.

**Training**  LayTextLLM is initialized with Llama2-7B-Base model, the pre-training, SFT, and other model hyper-parameters can be seen in Tab. 6. Please note that all variants of LayTextLLM, including those utilized in ablation studies, are trained in accordance with the SFT settings. All baseline results are sourced from their respective original papers, with the exception of the Llama2-7B series and the Llama2-7B$_{coor}$ series. These were re-implemented and can be referenced in [21, 23].

| | Backbone | Plora rank | Batch size | Max length | Precision | Train params | Fix params |
|---|---|---|---|---|---|---|---|
| **Pretrain** | Llama2-7B-base | 256 | 128 | 2048 | bf16 | 648 M | 6.7 B |
| **SFT** | Llama2-7B-base | 256 | 256 | 4096 | bf16 | 7.4 B | 0B |
| | **Learning rate** | **Weight decay** | **Scheduler** | **Adam betas** | **Adam epsilon** | **Warm up** | **Epoch** |
| **Pretrain** | 1.0e-04 | 0.01 | cosine | [0.9, 0.999] | 1.0e-08 | 0.005 | 2 |
| **SFT** | 2.0e-05 | 0.01 | cosine | [0.9, 0.999] | 1.0e-08 | 0.005 | 2 |

Table 6: LayTextLLM trainng Hyper-parameters.

**Inference**   For the document-oriented VQA test set, we use the original question-answer pairs as the prompt and ground truth, respectively. For Key Information Extraction (KIE) tasks, we reformat the key-value pairs into a question-answer format, as described in [12, 19, 21]. Additionally, for the FUNSD dataset, we focus our testing on the entity linking annotations as described in [21].

To eliminate the impact of randomness on evaluation, no sampling methods are employed during testing for any of the models. Instead, beam search with a beam size of 1 is used for generation across all models. Additionally, the maximum number of new tokens is set to 512, while the maximum number of input tokens is set to 4096.

## C   Discussion of Input Sequence Length

As mentioned in Section 4.6, it is intriguing that LayTextLLM has fewer input sequences than Do-cLLM, which is counterintuitive given that LayTextLLM interleaves bounding box tokens, typically resulting in longer sequence lengths. We attribute this to the Byte Pair Encoding (BPE) tokenizers [54] prevalently used in modern LLMs such as Llama2.

BPE operates by building a vocabulary of commonly occurring subwords (or token pieces) derived from the training data. Initially, it tokenizes the text at the character level and then progressively merges the most frequent adjacent pairs of characters or sequences. The objective is to strike a balance between minimizing vocabulary size and maximizing encoding efficiency.

Thus, when tokenizing a single word like *"International"* on its own, the tokenizer might identify it as a common sequence in the training data and encode it as a single token. This is especially likely if *"International"* frequently appears as a standalone word in the training contexts. However, when the word *"International"* is part of a larger sequence of words such as including in a long sequence of OCR-derived texts like *"...335 CPC,International,Inc..."*, the context changes. The tokenizer might split *"International"* into sub-tokens like *"Intern"* and *"ational"* because, in various contexts within the training data, these subwords might appear more frequently in different combinations or are more useful for the model to understand variations in meaning or syntax.

When using LayTextLLM, we input word-level OCR results into the tokenizer, typically resulting in the former situation, where words are encoded as single tokens. Conversely, with DocLLM, the entire OCR output is processed as one large sequence, leading to the latter situation and a longer sequence length than in LayTextLLM. This difference underscores the utility of LayTextLLM in achieving both accuracy and inference efficiency due to its shorter sequence length.

## D   Shuffle Ratio Exploration

Tab. 7 presents the results of exploring training and testing shuffling ratios on the FUNSD dataset using two different models: Llama2-7B-base and LayTextLLM. The table shows the performance of these models at various shuffling ratios (100%, 50%, 20%, and 0%).

LayTextLLM consistently outperforms Llama2-7B-base across all levels of shuffling, which further underscores the significance of interleaving spatial layouts with text. Particularly at the 100% shuffle level, Llama2-7B-base demonstrates limited accuracy at only 20.3, while LayTextLLM maintains a relatively higher performance. It is also interesting to note that Llama2-7B-base generally improves as the shuffling percentage decreases, whereas LayTextLLM performs best when 20% of the examples with OCR-derived text are shuffled. This observation suggests that LayTextLLM effectively utilizes spatial layouts and is less dependent on the sequence of input tokens. Therefore, a certain proportion of shuffled examples can serve as adversarial examples to enhance the model's robustness, addressing

situations such as errors in the text order from the OCR engine, which are caused by subtle differences in horizontal or vertical coordinates.

|  | FUNSD | |
|---|---|---|
| Ratio | Llama2-7B-base | LayTextLLM |
| 100% | 20.3 | 44.7 |
| 50% | 49.1 | 62.1 |
| 20% | 50.2 | **65.4** |
| 0% | 52.3 | 65.1 |

Table 7: Shuffling ratio exploration in FUNSD dataset.

# E  Results of ChartQA

As shown in Fig. 7, the question-answer pairs in ChartQA [43] tend to involve the visual cues for reasoning. However, with only text and layout information as input, the proposed LayTextLLM inevitably have difficulties in reasoning visual-related information. Thus, on the ChartQA dataset, LayTextLLM can hardly achieve better performance than previous methods that include visual inputs. Although the visual information is not used in LayTextLLM, it can still exhibit better zero-shot ability than UniDoc [10]. After incorporating the training set of ChartQA, the performance of LayTextLLM can be boosted to 35.7%. Considering the importance of visual cues in ChartQA-like tasks, we will try to involve the visual information into LayTextLLM in future work.

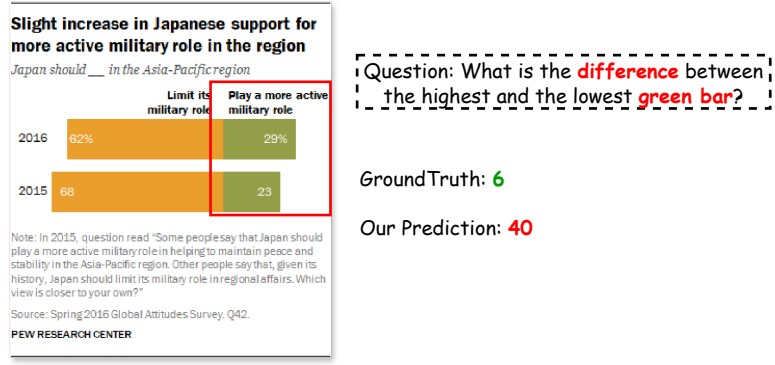

Figure 7: A failure case of LayTextLLM on CharQA.

|  | ChartQA |
|---|---|
| **OCR-free** | |
| UniDoc [10] | 10.9 |
| DocPedia [9] | 46.9* |
| Monkey [49] | 54.0* |
| InternVL [50] | 45.6* |
| InternLM-XComposer2 [15] | 51.6* |
| TextMonkey [12] | 58.2* |
| TextMonkey$_+$ [12] | **59.9*** |
| **text + polys** | |
| LayTextLLM$_{zero}$ (Ours) | 21.4 |
| LayTextLLM$_{vqa}$ (Ours) | 29.8* |
| LayTextLLM$_{all}$ (Ours) | 35.7* |

Table 8: Comparison with SOTA OCR-free MLLMs on ChartQA. * indicates the training set used.

