# OpenReview forum: "A Bounding Box is Worth One Token: Interleaving Layout and Text in a Large Language Model for Document Understanding"
_NeurIPS.cc/2024/Conference — Submitted to NeurIPS 2024_

### Official Review · Reviewer_tEb5 · 2024-06-19

**Soundness:** 2
**Presentation:** 3
**Contribution:** 3
**Rating:** 6
**Confidence:** 3

**Summary:**

This paper introduces the LayTextLLM method for document understanding, which encodes text positional information in the embedding space of an LLM and trains for effective understanding of document data as interleaved OCR-detected text and bounding box information. The results show improved performance compared to prior works on the KIE tasks, as well as on VQA in many cases.

**Strengths:**

The treatment of layout information as a modality interleaved with text is logical, and the use of a projection into the LLM’s embedding space to represent bounding boxes is clever and appears to be novel. The tasks approached are important and overall the proposed method does appear to improve document understanding (though this will be more convincing if the caveats listed below are addressed).

I also appreciate the focus on open-source models and data for the method and its evaluation, making the results reproducible.

**Weaknesses:**

There are some issues regarding the comparisons to existing models, making it unclear how much of the observed improvement is really due to the novel method proposed.

LayTexLLM is implemented with Llama-2-7b, but it seems that many models compared to (e.g. the strong-performing LayoutLLM) may use other LLM backbones, making it unclear whether the superior performance of LayTexLLM in many settings is due to the proposed novel method or the LLM backbone. The results will be more convincing with a comparison of different methods with the same LLM backbone (or at least an analysis of the number of parameters in each model).

It is not clear what OCR engine is used, raising the concern that different OCR engines could explain some of the gaps in performance between models being compared.

There are also issues with how the training is presented that make it difficult to interpret results. Some places (L131, L179, etc.) mention pre-training and SSFT, implying that pre-training means the LNTP training task. However, Sec 4.1 mentions “pre-training” and “SFT”, implying that pre-training refers to SSFT+LNTP and that it is followed by SFT (Supervised Fine Tuning) for particular tasks (VQA and KIE). The results also mention zero-shot and supervised results (e.g. L297), but it is unclear from the text and results tables which results are obtained zero-shot or from SFT, making it hard to understand if the comparisons are fair.

The statements about large improvements over SOTA MLMMs (L13-14, L83-84) seem slightly misleading since LayTextLLM uses OCR detections and thus is more comparable to other OCR-based methods.

LNTP (Sec. 3.2.1) is presented as a novelty but seems to just be the regular language modeling objective. If I understand correctly, this could be toned down to simply say that the added SLP and P-LoRA parameters are updated with a language modeling loss.

**Questions:**

I don’t fully follow the claim of L54-55 about autoregressive models vs. DocLLM-style models. Why would autoregressive modeling a priori be expected to outperform spatial cross attention for document understanding?

While the justification of SSFT (Sec 3.2.2) makes sense, it seems that the issue stems from the use of positional encodings which encode left-to-right order of tokens. Have you considered using positional encodings that directly encode (x, y)-positions of text to avoid this artifact or to give the model an inductive bias towards the layout’s 2D positioning?

Why was only Llama-2-7b used? Would the proposed method work for other LLM backbones?

What is the motivation for using P-LoRA (Sec. 3.1.2)? Is it applied to every layer?

L195 states that LNTP loss is calculated for text tokens only. Does this mean that bounding box tokens are still used as inputs but just not as targets? Why is this done, and is it tested?

There are a number of minor grammatical errors throughout the paper that need revision, including missing articles (e.g. L163 “to (the) visual modality”, among others) and some awkward wording (e.g. L142 “specific”, L160 “communicated”, L211 “cons and pros” => “advantages and disadvantages”, L262 “it’s” => “it is”, L332 “(and we provide further) discussion”, among others). The acronym SFT used throughout should be defined somewhere.

Tables 1-3: The term “polys” and the exact meaning of the asterisk * are unclear.

**Limitations:**

Limitations are clearly discussed in Section 5 (which should have the title “Limitations” in plural). Additionally, does the limitation of lacking visual cues apply to text formatting such as bolding or italics? This would connect well to the examples in Figure 6 where bold text is prominent.

---

> ### Author Rebuttal · Authors · 2024-08-03
>
> Thank you for your thoughtful feedback and appreciate the recognition of our paper’s contributions and novelty. We are grateful for the opportunity to address the concerns raised.
>
> **W1-Model backbone:** We implemented LayTexLLM using Llama2-7b, consistent with previous OCR-based methods like DocLLM, which also use Llama2-7b. We also replicated the results of the coor-as-tokens scheme using Llama2-7b for consistency. Noting the LayoutLLM model utilizes Llama2-7b and Vicuna 1.5 7B, which is fine-tuned from Llama2-7b. Therefore, for the majority of our comparisons, the models are based on the same or similar LLM backbones, allowing for a fair comparison between approaches.
>
> Other MLLMs use backbones like Qwen-VL, Internlm, and Vicuna, which are models with at least 7 billion parameters, excluding the visual encoder. Thus, we can say the comparison is fair, at least in terms of model parameters. We will explicitly mention this in the updated version.
>
> **W2-OCR engine**: We use word-level OCR from the respective datasets to ensure a fair comparison, except for the ChartQA dataset, where no OCR is provided (and we have mentioned this in lines 277-278). We will explicitly mention this in an updated version.
>
> **W3-Confusion of terminology**:
>
> Sorry for the confusion caused. Here are explanations:
>
>   - Pretrain and SFT Clarification: In Section 4.1, the terms "pretrain" and "SFT" refer to LNTP and SSFT. We will revise this section to avoid confusion.
>   - Zero-shot and Supervised Results: The term "zero-shot" refers to a model trained using SSFT only with Document Dense Description (DDD) and Layout-aware SFT data, as used in LayoutLLM. "Supervised" indicates that the model is trained using SSFT with DDD, Layout-aware SFT data, and the training sets of downstream datasets such as DocVQA and FUNSD. This terminology aligns with LayoutLLM, and we will clarify this in the updated version.
>   - Asterisk Notation: An asterisk (*) is used to indicate whether the corresponding training set of a downstream dataset is included in the training of a specific model. This notation facilitates a fair interpretation of experimental results for the reader.
>
> **W4-Statement about improvement**: We'll tone down the phrasing to accurately reflect this comparison and highlight our improvements in relation to OCR-based methods.
>
> **W5-LNTP**: We acknowledge that LNTP resembles the regular language modeling objective. We'll tone down the presentation to clarify that the added SLP and P-LoRA parameters are updated using standard language modeling loss.
>
> **Q1-claim of autoregressive vs. docllm**: We had a brief discussion in line 297 – line 299. Here we elaborated this in detail, and add them in the updated version:
>
>   - Disentangled Attention: DocLLM uses a disentangled attention to process spatial and textual modalities separately (using differerent QK weight) before integrating them. This independent handles of spatial information from document layouts, unlike traditional autoregressive models that process inputs sequentially (use the same suite of weights). In contrast, LayTextLLM interleaves bounding box tokens with text, unifying both modalities in a single sequence through an autoregressive approach.
>
>   - Block Infilling Objective: Unlike standard autoregressive models that predict the next token based only on preceding text, DocLLM uses block infilling to predict missing text blocks based on both preceding and succeeding context. This deviates from leveraging the inherent autoregressive nature of traditional LLMs which solely relies on preceding tokens.
>
>   - Impact on Performance: As demonstrated in Table 2. when compared using the same training dataset, LayTextLLM significantly outperforms DocLLM.
>
> **Q2-encode x and y**: We considered using positional encodings that directly encode (x, y) positions to address the artifact issue. However, to fully leverage the LLM parameters and maintain simplicity, we avoided encoding (x, y) positions, as it could complicate the model. Instead, we focused on balancing LLM reuse with necessary adjustments, which led us to propose SSFT.
>
> **Q3-model backbone and generalization of the method**: We implemented LayTexLLM using Llama2-7b as our LLM backbone, in line with prior OCR-based methods like DocLLM and LayoutLLM.
>
> Our method is model-agnostic. In our in-house KIE test, we evaluated the performance of the Baichuan2 7b and Qwen2 7b models. The results showed that incorporating the SLP layer improved performance for both models compared to not using it.
>
> | Model        | w/o SLP | With SLP |
> |--------------|---------|----------|
> | Baichuan2 7B | 0.7464  | **0.7738**   |
> | Qwen2 7B     | 0.754   | **0.7858**   |
>
> **Q4-using PLORA**: The motivation for using P-LoRA  is due to the concern of having too few learnable parameters.  P-LoRA is applied in each layer, but the main contribution still comes from SLP.
>
> **Q5-LNTP loss**: Yes, the bounding box tokens are used as inputs but not targets. Our objective is to understand the bounding boxes, not to generate them. Therefore, it is unnecessary to compute a loss for the bounding box tokens.
>
> Also, we tested in a in-house KIE dataset, finding that including bounding box as targets (using string like ''[1,20,10,30]'') during LNTP drops the downstream performance.
>
> **Q6-typo**: We will fix those typos in the updated version.
>
> **Q7-polys and asterisk**: The term 'polys' will be replaced with 'coordinates.' An asterisk (*) indicates if the training set of a downstream dataset is included in the training of a specific model. This notation ensures a fair interpretation of the experimental results for the reader.
>
> **Limitations**: Yes, bolding and italic texts should be included as visual cues, which will be updated in the updated version.
>
> We would appreciate it if you could improve your rating if all concerns been addressed and we look forward to your response.

---

> > ### Comment · Reviewer_tEb5 · 2024-08-12
> >
> > Thank you for your detailed and careful response. I believe this addresses my main concerns and I have updated my rating accordingly.
> > I encourage the authors to incorporate all of these clarifications into the final version, and particularly the points regarding fair comparisons between methods (LLM base models, OCR engines, ...).

---

> > > ### Author Response · Authors · 2024-08-12
> > > **Thanks**
> > >
> > > Thank you very much for your positive feedback and for updating your rating. We greatly appreciate your thoughtful and constructive comments. We are committed to incorporating all of the clarifications you suggested, particularly regarding the fair comparisons between methods, such as LLM base models and OCR engines, in the final version of our paper.

---

### Official Review · Reviewer_WMks · 2024-07-03

**Soundness:** 3
**Presentation:** 3
**Contribution:** 4
**Rating:** 7
**Confidence:** 4

**Summary:**

This work presents an innovative method for integrating layout information into LLMs to enhance document understanding tasks. Instead of treating bounding box coordinates as input text tokens, the bounding box information is embedded into a single token and interleaved with text tokens. This approach addresses the challenge of long sequences while leveraging the autoregressive nature of LLMs. Experimental results demonstrate the effectiveness of the proposed method, achieving state-of-the-art performance and resulting in shorter input sequence lengths.

**Strengths:**

1.Interleaving layout information and text is novel.

2.The proposed Shuffled-OCR Supervised Fine-tuning is interesting and may benefit other OCR-based approaches.

3.The approach achieves state-of-the-art performance on most text-rich VQA and KIE tasks, validating the effectiveness of interleaving layout and text and significantly reducing input length.

4.The paper is well-written, providing sufficient experimental details, ablations and discussions to comprehend each component of the model.

**Weaknesses:**

1.In layout-aware pretraining tasks, whether it is beneficial to predict both the bounding box and the text, rather than just the text.

2.LaytextLLM achieves satisfying performance in various tasks, but it will be better to incorporate the visual modality for more application scenarios.

**Questions:**

Is the input length shorter than the original input when not using a bounding box, only in the Llama tokenizer, or is this also observed in tokenizers of other LLMs?

**Limitations:**

Please refer to weaknesses.

---

> ### Author Rebuttal · Authors · 2024-08-03
>
> We thank the reviewer for the thoughtful feedback and appreciate the recognition of our paper’s contributions, writing and novelty. We are grateful for the opportunity to address the concerns raised.
>
> **W1-Compute loss of bounding box**:
>
>  - First of all, our objective is to understand the bounding boxes, not to generate them. Therefore, it is unnecessary to compute a loss for the bounding box tokens. Also, as shown in Figure 4, the prediction of t2 is made using the hidden state of b1, which means the supervised signal is backpropagated to the SLP.
>
>  - We tested including bounding boxes as targets (using strings like "[1,20,10,30]") during LNTP on an in-house KIE dataset and found that performance dropped.
>  - However, when we tested including bounding boxes as targets during SSFT, we found that the precision increased while the recall decreased, resulting in an almost unchanged micro F-score. Therefore, we can conclude that including bounding boxes as targets is beneficial only when added in the downstream tasks instead of during pretraining, and only when the application is sensitive to precision.
>
> **W2-Including visual modality**: We acknowledge that incorporating visual information can enhance performance, as discussed in the Limitations section. Exploring this further is a direction for our future research.
>
> **Q1-length reduction**:  The length reduction is universal and can be generalized to other LLMs. We have conducted additional tests on sequence lengths using the Baichuan2 tokenizer on an in-house KIE dataset, confirming the token reduction is universal and agnostic to LLMs used when compared to coor-as-tokens. When compared to DocLLM, we can ensure that LayTextLLM maintains an equal or shorter sequence length, regardless of the tokenizer used.
>
> |      Baichuan2 tokenizer     | LayTextLLM | DocLLM | Coor-as-tokens |
> |-----------|------------|--------|----------------|
> | Length| 313.27     | 313.27 | 1242.63        |

---

> > ### Comment · Reviewer_WMks · 2024-08-13
> >
> > I believe my concerns here are reasonably satisfied. I am impressed by the further discussion on the including bounding box as prediction target and the length reduction advantages brought by LayTextLLM, which I think is quite a nice addition to paper. Consequently, I keep the positive score and support the acceptance. I encourage the authors to incorporate these clarifications into the final version.

---

> ### Author Response · Authors · 2024-08-13
> **Thanks**
>
> Thank you very much for your valuable feedback and keeping the positive rating. We are committed to incorporating all of the clarifications you suggested, particularly regarding the discussion of bounding box prediction in the final version of our paper.

---

### Official Review · Reviewer_Nepi · 2024-07-11

**Soundness:** 3
**Presentation:** 3
**Contribution:** 2
**Rating:** 4
**Confidence:** 4

**Summary:**

The paper introduces a novel approach, named LayTextLLM, for document understanding tasks, which efficiently integrates spatial layouts and textual data within LLM. It employs a Spatial Layout Projector and introduces two innovative training tasks: Layout-aware Next Token Prediction and Shuffled-OCR Supervised Fine-tuning. Extensive experiments demonstrate significant improvements over previous state-of-the-art models in KIE and VQA. This paper demonstrates the importance of layout information in document understanding tasks.

**Strengths:**

1. The paper introduces a novel approach by integrating SLP and P-LoRA to effectively encode and process layout information. This method significantly improves the interaction between spatial layouts and textual data within LLM, providing a new direction for future research.
2. The paper proposes the LNTP task and SSFT task to enable the LLM to layout information, thereby enhancing its document understanding capabilities and improving performance on document-related tasks.

**Weaknesses:**

1. Due to miss the crucial visual information necessary for document understanding, this LayoutTextLLM heavily relies on OCR-derived text and spatial layouts. Other works such as LayoutLLM, layoutLMv3, introduces visual information to enhance the document understanding performance.
2. The exploration of the shuffling ratio was conducted only on Key Information Extraction (KIE) tasks. It should also be validated on Visual Question Answering (VQA) datasets to determine if the 20% shuffling ratio is optimal across different types of tasks.
3. The effectiveness of LNTP and SSFT methods should be substantiated with more ablation studies. It is recommended to fine-tune Llama2-7B directly using the existing data for a more comparisons.
4、Although LayTextLLM shows higher performance on DocVQA compared to LayoutLLM, this comparison is not entirely fair as LayoutLLM was evaluated in a zero-shot setting. Moreover, the zero-shot performance of LayoutLMv3 on DocVQA surpasses that of LayTextLLM.

**Questions:**

Compared with layoutlm series，what are the advantages of the encoding bounding box method proposed in this paper.

**Limitations:**

The author has already mentioned in the limitation section of the paper that the proposed model is difficult to handle scenarios where inference relies on visual cues.

---

> ### Author Rebuttal · Authors · 2024-08-03
>
> We thank the reviewer for the thoughtful feedback and appreciate the recognition of our paper’s novelty and improved performance. We are grateful for the opportunity to address the concerns raised.
>
> **W1-Lack visual modality**: We acknowledge that incorporating visual information can enhance performance, as discussed in the Limitations section. Exploring this further is a direction for our future research.
>
> **W2-Exploration of shuffle ratio**:
>
> We conducted an additional experiment using DocVQA, and the results again demonstrated the superiority of LayTextLLM, and confirming that choosing 20% is an appropriate selection. We will add this result in the updated version.
>
>
> | Ratio | **Funsd** |          |         | **DocVQA** |          |
> |-------|:---------:|:--------:|:-------:|:----------:|:--------:|
> |       | Llama2    | LaytextLLM |       | Llama2     | LaytextLLM |
> | 100   | 20.3      | 44.7     |       |       34.8     | 53.4    |
> | 50    | 49.1      | 62.1     |       |      63.1      | 72.8    |
> | 20    | 50.2      |**65.4**     |       |       64.7     | **73.4**   |
> | 0     | **52.3**      | 65.1     |       |        **65.5**    | 73.0    |
>
>
>
> **W3-More ablation studies**:
>
> We have conducted a new ablation study, which will be added in the next version. The experimental results demonstrating the interleaving of bounding box and text providing the largest boost, while LNTP+SSFT provide big improvement in VQA tasks.
>
> | SLP | P-LoRA | LNTP+SSFT | Document-Oriented VQA |               |               |               |         | KIE   |       |       |       |         |
> |-----|--------|-----------|-----------------------|---------------|---------------|---------------|---------|-------|-------|-------|-------|---------|
> |     |        |           | DocVQA                | InfoVQA       | VisualMRC     | Avg           | FUNSD   | CORD  | SROIE | POIE  | Avg     |
> |    |        |           | 71.5                  | 31.9          | 31.1          | 44.8          | 50.5    | 90.2  | 91.6  | 54.1  | 71.6    |
> | ✓   |      |           | 74.7                  | 35.7          | 32.5          | 47.6          | 55.1    | 94.9  | 94.6  | 68.3  | 78.2    |
> | ✓   | ✓      |          | 76.5                  | 38.0          | 30.6          | 48.4          | 54.3    | 95.9  | **95.3**  | **70.6**  | 79.0    |
> | ✓   | ✓      | ✓         | **78.8**              | **42.7**      | **34.4**      | **52.0**      | **63.0**| **95.9**| 95.2| 62.1| **79.1**  |
>
>
> Note the slight difference in the value from the previous version is due to our use of an in-house framework, while the new version is based on the Huggingface Transformers.
>
> **W4-fair comparison**
>
> - We use an asterisk (*) indicates whether the training set of a downstream dataset is included in the training of a specific model, which ensures a fair interpretation of the experimental results for readers. While we acknowledge that LayoutLLM performs better in the zero-shot DocVQA scenario, our primary comparison focuses on pure OCR layout + OCR text models, such as DocLLM and ICL-D3IE, as LayoutLLM incorporates the visual modality.
> - Could you please specify the citation that provides the zero-shot performance of LayoutLMv3, and we will include this information in the updated version.
>
> **Q1-advantage of projecting bounding box**
>
> - Precise Layout Representation: The LayoutLM series uses position embeddings for discrete layout representation, while LayTextLLM maps four coordinates into a continuous hidden space which is continuous. We believe this approach offers a more precise and enriched understanding of layout.
> - Enhanced Contextual Understanding: By interleaving spatial layout with textual content, the model enhances its understanding of context and structural relationships within documents. This is especially beneficial for layout-dependent documents such as invoices, forms, and multi-column scientific articles, and is particularly advantageous for decoder-only models like LLMs.
>
> We kindly request your acknowledgement of our reply, and are welcome to further discussions for your questions and concerns. We would be fully appreciated if you would consider to improve the **rating**. We look forward to your response.

---

> > ### Author Response · Authors · 2024-08-13
> > **Sincere Invitation to Participate in the Discussion**
> >
> > Dear Reviewer Nepi,
> >
> > We sincerely appreciate the time and effort you've dedicated to reviewing our work. As the discussion period is drawing to a close, we kindly request your acknowledgment of our reply. We value your insights and would be grateful if you could join the discussion to clarify any remaining questions or concerns. Your input is highly valued, and we would greatly appreciate it if you could consider improving the evaluation after reviewing our responses.
> >
> > Thank you very much for your consideration.
> >
> > Sincerely, The Authors

---

> > > ### Author Response · Authors · 2024-08-14
> > > **Gentle Follow-Up on Review Response Acknowledgment**
> > >
> > > Dear Reviewer Nepi,
> > >
> > > We understand that you have many commitments, and we deeply appreciate the time you've already devoted to reviewing our work. As the discussion phase is coming to an end, we kindly request your acknowledgment of our reply。
> > >
> > > We would be very grateful if you could acknowledge our response and share any further thoughts or clarifications you might have. Your feedback is incredibly valuable to us, and we sincerely hope that our responses have addressed your concerns.
> > >
> > > Thank you again for your time and consideration.
> > >
> > > Sincerely, The Authors

---

### Official Review · Reviewer_uyfs · 2024-07-13

**Soundness:** 2
**Presentation:** 3
**Contribution:** 1
**Rating:** 5
**Confidence:** 4

**Summary:**

This paper presents LayTextLLM, a novel approach to document understanding that effectively integrates spatial layout information and text into a large language model. Existing methods that integrate spatial layout with text often produce excessively long text sequences. LayTextLLM addresses these problems by projecting each bounding box into a single embedding and interleaving it with text.  The method is evaluated on Key Information Extraction (KIE) and Visual Question Answering (VQA) tasks.

**Strengths:**

- Effective sequence reduction: The proposed method reduces the length of text sequences, addressing a common problem in document understanding.
- Performance improvement: LayTextLLM demonstrates improvements in KIE and VQA tasks, showing performance gains over alll state-of-the-art models.
- Evaluation: The paper provides detailed benchmark evaluations on 2 tasks and 7 datasets

**Weaknesses:**

- Incomplete related work: The paper omits several relevant OCR-based models, such as UDOC, LayoutMask, BROS, LAMBERT, DocFormer and LiLt.
- Insufficient explanation: The repeated claim that DocLLM cannot fully exploit autoregressive features is not adequately explained.
- Limited comparisons: There is no comparison with alternative methods that embed coordinates, such as co-as-token approaches (Lmdx, Shikra, ICL-D3IE).
- Marginal token reduction: The reduction in the number of tokens appears to be limited, and the paper does not clarify whether words or lines are encoded, which could have a significant impact on token reduction.

**Questions:**

- Why use an embedding to encode coordinates that are already 4D vectors? What is the gain, considering there is no additional information (e.g., font, style, zone type)?
- How can SLP be trained if there is no loss computed on bounding box tokens?
- How do you explain the good performance of LayTextLLM_zero?

**Limitations:**

- Limited comparisons: The paper primarily compares LayTextLLM to DocLLM, which may not provide a comprehensive assessment of its performance.
- Impact of token reduction: The reduction in the number of tokens, while beneficial, appears to be limited and may not provide significant practical benefits in all scenarios.

---

> ### Author Rebuttal · Authors · 2024-08-02
>
> Thank you for taking time to review our paper.  We thank the reviewer for the thoughtful feedback and are grateful for the opportunity to address the concerns raised.
>
> **W1-Incomplete related work**: Our survey focuses on decoder-only architectures within LLMs to highlight their unique capabilities. We acknowledge the need for more comprehensive coverage and will include citations on encoder-only architectures in the updated version.
>
> **W2-Insufficient explanation**:We had a brief discussion in line 297 – line 299. Here we elaborated this in detail, and add them in the updated version:
>
>   - Disentangled Attention Mechanism: DocLLM introduces a disentangled attention mechanism that processes spatial and textual modalities separately. This mechanism handles spatial information independently before merging it (using different attention QK weight)  with textual information, which is different from traditional autoregressive models that process inputs sequentially without such separation (using the same suite of weights). While LayTextLLM introducing bounding box token interleaving with texts, unifying bounding box and text information in a single sequence, fully fusing these two modalities by autoregressive method.
>
>   - Block Infilling Objective: Unlike standard autoregressive models that predict the next token based on the preceding sequence, DocLLM uses a block infilling approach where it predicts missing text blocks based on both preceding and succeeding context. This deviates from leveraging the inherent autoregressive nature of traditional LLMs which solely relies on preceding tokens.
>
>   - Impact on Model's Predictive Performance: The experimental findings highlight the superiority of our method, LayTextLLM, as demonstrated in Table 2. When compared using the same training dataset, LayTextLLM significantly outperforms DocLLM.
>
> **W3-Limited comparison**
> - ICL-D3IE: We have included a comparison with ICL-D3IE. The data in Figure 1 and Table 2 is sourced from the ICL-D3IE paper (Coord-as-tokens-ICL-175B(Davinci-003), Table 2 Davinci-003-175Bcoor). We also replicated the ICL-D3IE using LLama2-7b (Figure 1, Coord-as-tokens-7B(Llama2), Table 2 Llama2-7B-chatcoor). A detailed discussion of the comparison with ICL-D3IE can be found in lines 301-307. In the updated version, we will change the term "coor-as-tokens" to "ICL-D3IE."
>  - Lmdx: Our comparison primarily focuses on approaches based on open-source LLMs instead of proprietary ones, as noted by Reviewer teB5. This ensures our results are reproducible. Therefore, LMDX was not included. Additionally, LMDX only provides results for CORD, whereas our experiments cover a broader range of text-rich VQA and KIE datasets. However, we will include a comparison with LMDX in the updated version.
>  -  Shikra: Shikra is not a document AI LLM even without proper OCR ability and is therefore outside the scope of this comparison. As noted by Reviewer teB5, the primary comparison should be with other OCR-based methods, such as DocLLM.
>
> **W4-Marginal token reduction**
>  - We utilize word-level OCR from the corresponding datasets to ensure a fair comparison, which will be explicitly mentioned in a later version of the document.
>
>  - Our claim of significant token reduction is primarily focused on the comparison with the coor-as-tokens scheme, as detailed in lines 74-77. In these instances, the reduction in tokens is substantial rather than marginal and applies to both word-level and line-level OCR. For example, when using the coord-as-tokens scheme with the Llama2 tokenizer, the coordinate string "[70,73,90,77]" occupies 13 tokens, while LayTextLLM represents the same information with just 1 token.
>
> |      Baichuan2-tokenizer     | LayTextLLM | DocLLM | Coor-as-tokens |
> |-----------|------------|--------|----------------|
> | Length| 313.27     | 313.27 | 1242.63        |
>
>  - Furthermore, compared to DocLLM, our approach yields either shorter or equivalent sequence lengths. We have conducted additional tests on sequence lengths using the Baichuan2 tokenizer on an in-house KIE dataset, confirming the token reduction is universal and agnostic to LLMs used.
>
>
> **Q1-encoding vector:** Using an embedding to encode coordinates is not about introducing additional information. Instead, it is a practice about transforming the coordinates into a hidden state that is more understandable for a LLM. This process involves aligning dimensions in a way that is more suitable for the model's architecture. For example, in LLAVA, the 1024-dimensional output from CLIP is mapped to 4096 dimensions to better align visual and text modality.
>
> **Q2-Train SLP:** There seems to be a misunderstanding regarding the training process. Although the loss is not computed for the bounding box token, the SLP is still trained. For instance, in Figure 4, the prediction of t2 is made using the hidden state of b1, which means the supervised signal is backpropagated to the SLP.
> Furthermore, our objective is to understand the bounding boxes, not to generate them. Thus, it is unnecessary to compute a loss for the bounding box tokens.
>
> **Q3-performance of LaytextLLM_zero:** The strong performance is primarily due to the interleaving of bounding box tokens with text, along with other design elements such as LNTP and SSFT. Additionally, the use of synthetic data from LayoutLLM, including DDD and layout-aware SFT, also contributes to training LayTextLLM_zero.
>
> We kindly request your acknowledgement of our reply, and are welcome to further discussions for your questions and concerns. We would be fully appreciated if you would consider to improve the **rating**. We look forward to your response.

---

> > ### Author Response · Authors · 2024-08-13
> > **Sincere Invitation to Participate in the Discussion**
> >
> > Dear Reviewer uyfs,
> >
> > We sincerely appreciate the time and effort you've dedicated to reviewing our work. As the discussion period is drawing to a close, we kindly request your acknowledgment of our reply. We value your insights and would be grateful if you could join the discussion to clarify any remaining questions or concerns. Your input is highly valued, and we would greatly appreciate it if you could consider improving the evaluation after reviewing our responses.
> >
> > Thank you very much for your consideration.
> >
> > Sincerely, The Authors

---

> > ### Comment · Reviewer_uyfs · 2024-08-13
> >
> > Thank you for your answers, which have clarified a number of points. I'm going to raise my score.

---

> > > ### Author Response · Authors · 2024-08-14
> > > **Thanks**
> > >
> > > Thank you very much for the positive feedback and for updating your rating. We greatly appreciate your thoughtful and constructive comments. We are committed to incorporating all of the clarifications you suggested in the final version of our paper.

---

### Author Response · Authors · 2024-08-11
**General Response: Invitation to Join in Discussion**

Dear Reviewers,

We sincerely appreciate the time and effort you've dedicated to reviewing our work. We are grateful for the recognition of the novelty and performance improvements acknowledged by all the reviewers. Due to the rush in finalizing the writing, some aspects may have caused confusion and misunderstanding. We have addressed these issues in our rebuttal and provided further elaboration.

As the discussion period is drawing to a close, we kindly request your acknowledgment of our reply. We are open to further discussions to clarify any remaining questions or concerns. We would greatly appreciate it if you could consider improving the evaluation after reviewing our responses.

Thank you very much for your consideration.

Sincerely,
The Authors

---

### Decision · Program_Chairs · 2024-09-25

**Decision:**

Reject

**Comment:**

The paper proposes interleaving spatial layouts with textual data through embeddings within a Large Language Model (LLM). The experiments demonstrate enhancements in Key Information Extraction (KIE) tasks and Visual Question Answering (VQA) tasks. We appreciate the responses from the authors and the discussions with reviewers.

However, the work could be enhanced by addressing the following points:
1) The paper should include more detailed discussions and analyses concerning the introduction of layout embedding (SLP). It is crucial to delve deeper into why this embedding significantly boosts performance. For instance, questions about the robustness of the embedding representation derived from layout coordinates should be explored. Given that document layouts can vary significantly across different types, and even within the same type of documents, it is important to assess whether changes in layout (e.g., scale) impact performance.
2) The potential influence of OCR results on the comparative analysis needs further examination as also pointed out by reviewers. The discussion on fair comparisons in the experimental section should be expanded. It is necessary to conduct more thorough discussions and analyses to determine whether these improvements are attributable to differences in base models or if they stem from the inclusion of "layout coordinates" as a critical feature in these tasks.